# Exploring chilling stress and recovery dynamics in C4 perennial grass of *Miscanthus sinensis*

Karolina Sobańska[1]☯*, Monika Mokrzycka[1]☯, Martyna Przewoźnik[1], Tomasz Pniewski[1], Katarzyna Głowacka[1,2]*

**1** Polish Academy of Sciences, Institute of Plant Genetics, Poznan, Poland, **2** Department of Biochemistry, University of Nebraska-Lincoln, Lincoln, NE, United States of America

☯ These authors contributed equally to this work.
* ksob@igr.poznan.pl (KS); kglowacka2@unl.edu (KG)

**Data Availability Statement:** All relevant data are within the manuscript and its Supporting Information files.

## Abstract

The increasing cultivation of perennial C4 grass known as *Miscanthus spp.* for biomass production holds promise as a sustainable source of renewable energy. Unlike the sterile triploid hybrid of *M. × giganteus*, which cannot reproduce through seeds, *M. sinensis* possesses attributes that could potentially address these limitations by effectively establishing itself through seed propagation. This study aimed to investigate how 18 genotypes of *M. sinensis* respond to chilling stress and subsequent recovery. Various traits were measured, including growth and biomass yield, the rate of leaf elongation, and a variety of chlorophyll fluorescence parameters, as well as chlorophyll content estimated using the SPAD method. Principal Component Analysis revealed unique genotype responses to chilling stress, with distinct clusters emerging during the recovery phase. Strong, positive correlations were identified between biomass content and yield-related traits, particularly leaf length. Leaf growth analysis delineated two subsets of genotypes: those maintaining growth and those exhibiting significant reductions under chilling conditions. The Comprehensive Total Chill Stress Response Index (SRI) pinpointed highly tolerant genotypes such as Ms16, Ms14, and Ms9, while Ms12 showed relatively lower tolerance.

## Introduction

The increasing cultivation of *Miscanthus* spp. for biomass production throughout Europe underscores its potential as a renewable energy resource. This perennial C4 grass is notable for its superior efficiency in utilizing nitrogen, water, and light energy, outperforming C3 plants [1–4]. Moreover, the establishment of *Miscanthus* spp. plantations contributes to enhancing soil carbon sequestration, further enhancing its attractiveness for environmentally sustainable biomass production [5, 6]. Given these attributes, *Miscanthus* spp. emerges as a promising candidate for large-scale, eco-friendly biomass generation. The hybrid *Miscanthus × giganteus* has garnered significant attention in the quest for a bioenergy crop [7]. However, several obstacles, including the high costs of mass propagation due to its sterile allopolyploid nature, and

**Funding:** This work was supported by the National Center Science Poland (NCN) [grant 2018/29/N/NZ9/00854].

**Competing interests:** The authors have declared that no competing interests exist.

**Abbreviations:** $PI_{ABS}$, performance index for PSII; $F_v/F_m$, maximum quantum yield of PSII; $F_0/F_m$, quantum yield of energy dissipation; $F_v/F_0$, potential activity of PSII; LER, leaf elongation rate; SRI, total chilling stress response index.

breeding challenges due to its triploid genome, have led to consideration of its progenitor, *Miscanthus sinensis*, as a more viable alternative. *M. sinensis*, capable of sexual reproduction through seeds and viable establishment, possesses attributes that could potentially overcome the limitations associated with *M. × giganteus* [8]. Investigating the genetic and agronomic equivalence of *M. sinensis* to *M. × giganteus* could facilitate its utilization as a bioenergy crop and provide a solution for the broader adaptation of *Miscanthus* spp. to diverse environmental conditions [9].

Beyond agronomy, understanding the stress tolerance mechanisms of *M. sinensis* is crucial for assessing its growth potential across various environmental scenarios and for potentially improving *M. × giganteus* [8]. Most C4 species face constraints in carbon assimilation under cooler temperatures (below 15°C) [10]. Cold stress can be categorized into chilling stress and frost stress, with chilling stress referring to temperatures below optimal growth but above 0°C. Nonetheless, certain *Miscanthus* accessions stand out as exceptions, exhibiting chilling tolerance by sustaining photosynthetic capacity within the range of 10°C to 16°C, akin to C3 species [11]. Multiple *Miscanthus* accessions demonstrate superior chilling stress tolerance compared to other C4 grasses such as maize, sorghum, or sugarcane [12–15]. However, variability in this trait is evident within the *Miscanthus* spp. germplasm pool [12, 16–18]. The intriguing phylogenetic relationship between *Miscanthus* spp. and cold-sensitive crops, such as maize or sugarcane, underscores the importance of exploring the potential correlation between chilling tolerance and biomass yields [12, 19, 20]. The primary practical concern revolves around whether enhancing the chilling tolerance of *Miscanthus* spp. can lead to higher biomass yields. Theoretically, the ability of *Miscanthus* spp. to thrive in colder conditions and produce shoots results in a longer canopy duration, allowing for more sunlight absorption and potentially increased biomass production. Despite having a lower photosynthetic capacity compared to maize, *Miscanthus* spp.'s larger leaf area and extended growth season enable greater biomass accumulation [3]. The length of the growing season, crucial for biomass production, is determined by the period between the last spring frost and the first autumn or winter frost. Genotypes that can thrive in low early spring temperatures and achieve early canopy closure absorb more solar radiation within these limits [16, 21]. Chilling stress, resulting from suboptimal temperatures, poses a significant challenge to the cultivation of *M. sinensis*, particularly in temperate climates. Investigating chilling stress and recovery dynamics in *M. sinensis* not only deepens our understanding of its physiological responses but also contributes to optimizing cultivation practices and enhancing breeding programs. This research involves selecting stress-tolerant genotypes and developing new varieties through methods such as interspecies crosses, genotype selection by natural processes, or interspecific hybridization.

In this study, we aimed to address pivotal questions: How do the responses of the diverse array of 18 genotypes of *M. sinensis* manifest under varying temperature conditions during the initial growth phases within controlled environments? What distinct physiological attributes delineate the chilling-tolerant genotype from the chilling-sensitive ones? This study delved comprehensively into the intricate interplay of physiological processes and a diverse array of traits that underlie biomass production. This meticulous exploration of *Miscanthus* spp. variants underscores the complex correlations linking physiological responses, temperature-induced stress, and recovery, offering an invaluable perspective to unlock their potential as sources of chilling genotypes for biomass production.

## Materials and methods

### Chilling stress experiment design

In Central and Eastern Europe, there is a notable occurrence of ground temperatures dropping significantly in early spring, particularly during the night or early morning hours. This

phenomenon entails temperatures dipping around 0˚C, while remaining above freezing during the daytime. Such conditions have a noteworthy influence on the subsequent growth and development of young plants. Hence, our motivation to delve into and delineate this phenomenon and its ramifications within a multi-year collection of grasses belonging to the *Miscanthus* spp. The chilling stress response was studied on 18 diploid *Miscanthus sinensis* genotypes (Table 1) from the multiannual collection of the Institute of Plant Genetics, Polish Academy of Sciences (IPG PAS), Poland. Each experimental pot (50L container) was filled with a 1:1 mixture of peat and sand, with a single 5cm-long rhizome piece placed in each. Soil nutrient levels were regulated by introducing a nutrient solution containing 100 mg/kg N, 50 mg/kg P, 200 mg/kg K, 2 mg/kg Fe, 10 mg/kg Zn, and 10 mg/kg Ca. Combined with the reserves stored in the rhizomes, this provided sufficient nutrients for the entire experiment duration. Plants were watered daily to field capacity throughout the experiment. They were grown in a greenhouse under white fluorescent tubes, providing a 12-hour photoperiod of photosynthetically active radiation at 750 µmol m$^{-2}$ s$^{-1}$ light intensity at canopy level, with temperatures maintained at 25˚C during the day and 19˚C at night, and relative humidity at 60% ± 10%. The experiment was performed under 750 µmol m$^{-2}$ s$^{-1}$ of light to mimic the light conditions which field-grown plants might experience in spring and early fall when chilling events occur. Plantlets of each genotype were cultivated in greenhouses until they reached the five-leaf stage. Following this stage, 10 plantlets from each genotype were transferred to a phytotron at the Wielkopolska Centre for Advanced Technologies in Poznan [22]. The conditions in the

**Table 1. List of the 18 *Miscanthus* spp. genotypes.**

| Name | Code | Accession identifier | Source |
|---|---|---|---|
| *Miscanthus sinensis* hybrid (2x) | Ms1 | 92M 012 1012 | Denmark |
| | Ms5 | 92M 017 9017 | Not known |
| | Ms6 | 92M 017 9020 | Not known |
| | Ms7 | 93M 000 7016 | Jelitto, Schwarmstedt |
| | Ms8 | 93M 000 7048 | Jelitto, Schwarmstedt |
| | Ms9 | 93M 014 6002 | Not known |
| | Ms10 | 93M 014 6010 | Mother of *Miscanthus sinensis* Grosse fontane, crossed with other *Miscanthus sinensis* |
| | Ms11 | 93M 014 6026 | Not known |
| | Ms12 | 93M 014 7009 | Not known |
| | Ms13 | 93M 014 8006 | Mother of *Miscanthus sinensis* Grosse fontane, crossed with other *Miscanthus sinensis* |
| | Ms14 | 93M 015 0013 | Mother of *Miscanthus sinensis* form Botanic Garden in Dresden, crossed with other *Miscanthus sinensis* |
| | Ms15 | 93M 015 0015 | Mother of *Miscanthus sinensis* form Botanic Garden in Dresden, crossed with other *Miscanthus sinensis* |
| *Miscanthus sinensis* From Japan | Ms16 | 93M 000 6004 | Mt. Hikagedaira, 1400 m NN, Gifu Prefecture |
| | Ms17 | 93M 000 6002 | Mt. Hikagedaira, 1400 m NN, Gifu Prefecture |
| | Ms18 | 93M 000 6006 | Mt. Hikagedaira, 1400 m NN, Gifu Prefecture |
| *Miscanthus sinensis* cv. Goliath | Ms19 | 92M 0088 | Pagels, Leer |
| *Miscanthus sinensis* cv. Grosse fontane | Ms20 | 92M 0086 | Pagels, Leer |
| *Miscanthus sinensis* cv. Silberfeder | Ms21 | 92M 0039 | Botanic Garden, Liverpool |

phytotron were carefully controlled to mimic those of the greenhouse environment. White fluorescent tubes provided a 12-hour photoperiod, delivering photosynthetically active radiation at an intensity of 750 µmol $m^{-2}$ $s^{-1}$ at canopy level. Daytime temperatures were maintained at 25˚C, while nighttime temperatures were set at 19˚C. Relative humidity was kept at 65% ± 10%. After 10 days of acclimation to the controlled-environment chamber, 5 out of 10 plants of each genotype were randomly selected and transferred to a second identical controlled environment chamber with pre-hardening settings: 7 days at 10˚C/5˚C day/night, followed by hardening at 5˚C/2˚C day/night for 14 days, before exposure to chilling extremes of 0.5˚C/0˚C for 24 hours on the 22nd day of the experiment. Subsequently, plants were allowed to recover at a constant temperature of 25˚C for 7 days, concluding on the 29th day of the experiment. Control plants were grown under the same conditions in the controlled-environment chamber at a constant temperature of 25˚C, with other environmental conditions unchanged. To prevent confounding genotypes with any undetected variation, the position of each plant within the chamber was randomized every second day.

For ease of explanation, chilling-treated plants and control plants were divided into groups labeled "T" and "TR" respectively. "T" denoted chilling-treated plants subjected to chilling stress, followed by recovery ("TR"). Control plants that grew continuously at 25˚C were labeled "C", and those compared to chilling-treated plants during recovery were labeled "CR". Importantly, plants labeled "T" and "C", as well as "TR" and "CR", were of the same age (Fig 1A).

## Leaf elongation

Leaf elongation data were collected from the uppermost leaf on the stem, where the ligule had not yet emerged from the sheath of the preceding leaf (i.e., growing leaves). The distance from the top edge of the pot to the tip of the selected leaf was measured using a ruler [1cm ± 1mm]. Leaf growth for each plant was calculated by subtracting the measurement on day zero from subsequent measurements. Developing leaves were measured every seven days during the pre-hardening, hardening, chilling extremum, and corresponding control treatments (Fig 1A).

## Biometric measurements

Biometric measurements, such as shoot height, length, and width (defined as the distance between the furthest points on the blade edge, measured perpendicular to the axis running from the leaf's tip to its base) of the first fully developed leaf (the youngest, i.e., the uppermost leaf with a ligule), stem diameter, and number of leaves per clump, were measured on the last day of chilling treatment (day 22nd of the experiment) and on the seventh day of the recovery phase (day 29th of the experiment), with data collected at the same time points for plants growing under control conditions (Fig 1A).

## Fluorescence and spectrometric measurements

Fluorescence and spectrometric measurements were conducted on leaf fragments. These measurements were taken along the longest distance between any two points on the blade's edge, perpendicular to the axis running from the tip to the base of the first fully expanded youngest leaf, which is identified as the uppermost leaf with a ligule. The relative chlorophyll content was determined in the morning using the SPAD method, which relies on absorbance at 650 nm and 940 nm, utilizing the SPAD 502 Plus/502 DL Plus instrument from Konica Minolta, INC., Japan.

Chlorophyll fluorescence was assessed using a fluorometer (Multi-Function Plant Efficiency Analyzer, Hansatech, Pentney, UK). Prior to measurements, each middle portion of the first completely expanded leaf was dark-adapted for 30 minutes. The measurement protocol

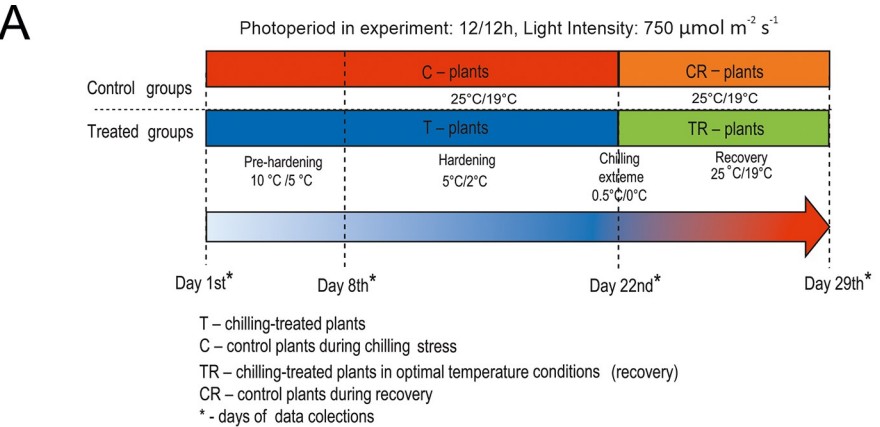

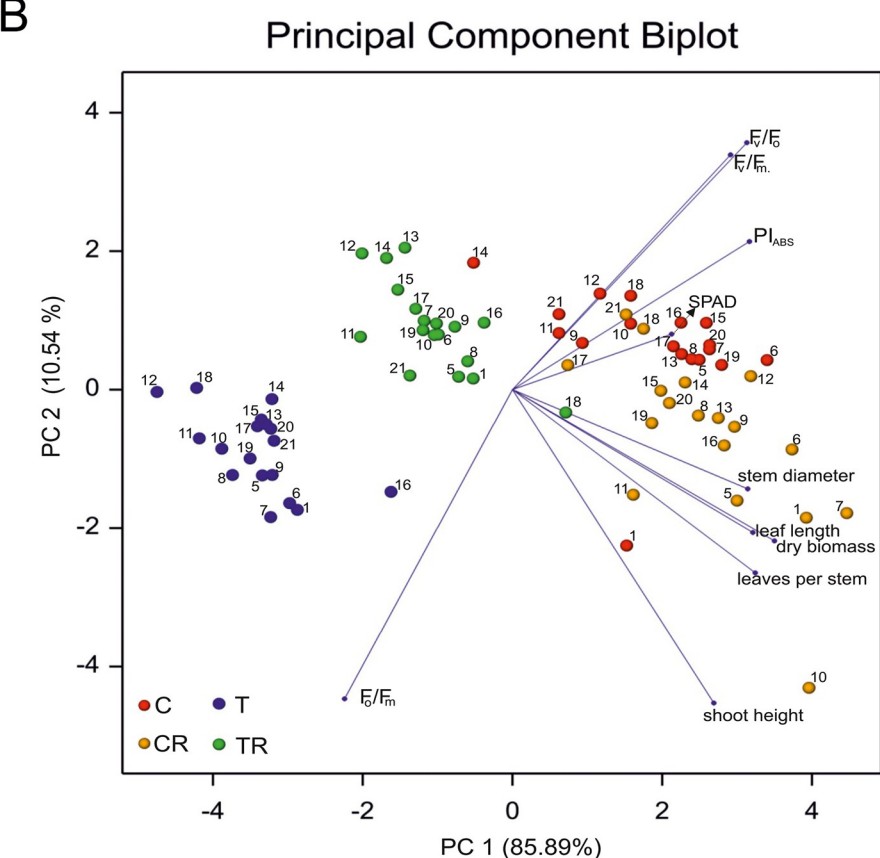

**Fig 1. A.** Graphical representation of the chilling/cold treatment experiment; **B.** Principal component biplot of 18 *M. sinensis* genotypes under chilling experiment. Dots represent the genotypes (coded by numbers, e.g., 1—Ms1) observed in chilling stress treatment (T; blue), control conditions (C; red), and recovery from chilling stress treatment (TR; green) and corresponding control recovery treatment (CR; orange). Vectors represent observed traits.

involved a light pulse with an intensity of 3500 μmol (photon) $m^{-2} s^{-1}$, lasting for 1 second, at a wavelength of 635 ± 10 nm. Some biophysical parameters relevant to this study include: $PI_{ABS}$ (performance index for PSII), which describes the initial phase of photosynthetic activity in an RC complex, involving three functional stages: light energy absorption (ABS), excitation

energy trapping (TR), and excitation energy conversion into electron transport (ET). $PI_{ABS}$ is calculated as: $PI_{ABS}$ = (RC/ABS) · (TR0/DI0) [ET0/(TR0 –ET0)]; $F_v/F_m$ (maximum quantum yield of PSII); $F_o/F_m$ (quantum yield of energy dissipation); $F_v/F_o$ (a ratio between the rate constants of photochemical and non-photochemical quenching of excited chlorophyll molecules). The data obtained were processed for the calculation of photosynthetic parameters using PEA Plus software version 1.12 (Hansatech).

Data collection occurred on the 22nd day of the experiment after the conclusion of the chilling treatment, and on the seventh day of the recovery phase (day 29th of the experiment), with corresponding measurements taken for plants growing under control conditions (Fig 1A).

## Statistical analyses

The data underwent analysis of variance (ANOVA) within a linear model, with fixed effects of genotype and treatment, as well as the interaction effect between genotype and treatment. In the two-way ANOVA, we are modelling considered trait as a function of genotype and treatment and interaction between them using R software. A post hoc TukeyHSD test was then performed to examine significant differences (significance level $\alpha = 0.05$) among pairs of sample means for all treatment conditions. For each genotype, Welch's T-test was employed with $\alpha = 0.05$ to test significant differences for control and treatment, and for recovery control and treatment. Principal component analysis (PCA) was carried out and visualized as a biplot using Genstat 21 [23] after standardizing the data. Leaf elongation rate (LER) was determined as the regression coefficient of leaf length over time, similar to the method described by Głowacka et al. [12] utilizing the lm function in R software. Differences between the mean control and treatment LERs for each genotype were assessed using Welch's T-test with a significance level of 0.05. The individual chill stress response index (ICSRI) was computed as the ratio of the parameter value under treatment conditions ($P^T$) to the parameter value under control conditions ($P^C$), following the approach outlined by [24]:

$$ICSRI = \sum\nolimits_{i=1}^{10} \frac{P_i^T}{P_i^C}.$$

The ICSRI was calculated for both chilling stress data and recovery conditions. Based on individual indices, combined treatment chill stress response index and combined recovery chill stress response index were computed as the sum of individual indices for each parameter. The total chill stress response index (TCSRI) was obtained by summing both combined indices. Lower values of TCRSI indicate sensitivity to chill genotypes, while higher values suggest high chill-tolerant genotypes. The stress susceptibility index (SSI) was computed for each feature in both chilling stress and recovery conditions as the ratio of the difference of one and the ratio of the parameter value at treatment and control conditions, divided by the stress intensity. The stress intensity is equal to one minus the ratio of the mean value of the parameter at treatment conditions to the mean value of the parameter at control conditions [25]:

$$SSI = \frac{1 - \frac{P^T}{P^C}}{SI}$$

where $SI = 1 - \frac{mean(P^T)}{mean(P^C)}$.

A smaller value of SSI indicates greater stress tolerance. Mean stress susceptibility indices for both treatment and recovery data were calculated, followed by the computation of the mean SSI for all conditions. The relationship between two variables in a data set was computed by Pearson method and the correlation matrix is visualized using R software.

## Results

### Variability in genotypic response to chilling stress and recovery

The investigation aimed to analyze differential responses among genotypes under distinct conditions, including the control (C), chilling stress treatment (T), and control and treatment recovery (CR and TR, respectively), using Principal Component Analysis (PCA) (Fig 1B). This analysis revealed significant trends where various biometric traits, such as leaf length, stem diameter, shoot height, leaves per stem, dry biomass yield, SPAD value, and various chlorophyll fluorescence parameters ($F_v/F_o$, $F_v/F_m$, $PI_{ABS}$), showed contrasting reactions concerning the chlorophyll fluorescence parameter quantum yield of energy dissipation ($F_o/F_m$). The results are also confirmed by the correlation matrix (S1 Fig).

In the control group (not subjected to chilling; Fig 1B), there was natural variability, with distinct clusters, for chilling treatment and recovery along PCA1 and PC2, and control and control recovery among PC2. Within the recovery and control treatments, there was more internal variability, particularly noticeable in genotypes like Ms17, Ms21, Ms18, Ms10, Ms1, Ms7, Ms5, Ms6, and Ms11, which displayed heightened phenotypic plasticity. All 18 genotypes diverged from control conditions under chilling stress (T), indicating their sensitivity to it (Fig 1B). The recovery treatment (TR) formed a cluster closer to the control groups. Genotypes Ms12, Ms13, Ms14, and Ms18 showed increased phenotypic plasticity under chilling stress (Fig 1B). Ms14's response during chilling and recovery phases closely matched its control phase, suggesting consistent behavior across phases for analyzed traits. Yield-related traits, such as stem diameter, leaf length, and leaves per stem, were correlated with dry biomass content, with leaf per stem exhibiting the strongest correlation (Fig 1B and S1 Fig). Chlorophyll fluorescence parameters showed strong correlations between $F_v/F_m$ and $F_v/F_o$, and medium correlation between $PI_{ABS}$ and SPAD values (Fig 1B and S1 Fig).

### Analysis of variance

The results of the analysis of variance (ANOVA) indicated significant effects of treatment, genotype, and the treatment × genotype interaction on almost all traits, except for the genotype effect on $F_v/F_o$ and the interaction of dry biomass (Table 2). Subsequent post hoc TukeyHSD test revealed that the differences in pairs of sample means for control and treatment, as well as for recovery control and treatment, were significant (p-value < 0.001) for all traits.

### Growth and biomass yield under chilling and recovery treatments

Various growth parameters and biomass yield, including leaf length (Fig 2), number of leaves per stem (S2 Fig), stem diameter (S3 Fig), shoot height (S4 Fig), and dry biomass (Fig 3),

**Table 2. Results of analysis of variance for observed traits (p-values for testing significance of variation sources).**

| trait | genotype | treatment | interaction |
|---|---|---|---|
| shoot height | <0.001 | <0.001 | <0.001 |
| stem diameter | <0.001 | <0.001 | 0.010 |
| leaf length | <0.001 | <0.001 | <0.001 |
| number of leaves per stem | <0.001 | <0.001 | 0.002 |
| dry biomass | 0.028 | <0.001 | 0.761 |
| SPAD-VALUE | <0.001 | <0.001 | 0.001 |
| $F_v/F_m$ | 0.016 | <0.001 | 0.003 |
| $F_o/F_m$ | <0.001 | <0.001 | <0.001 |
| $F_v/F_o$ | 0.052 | <0.001 | 0.039 |
| $PI_{ABS}$ | 0.008 | <0.001 | <0.001 |

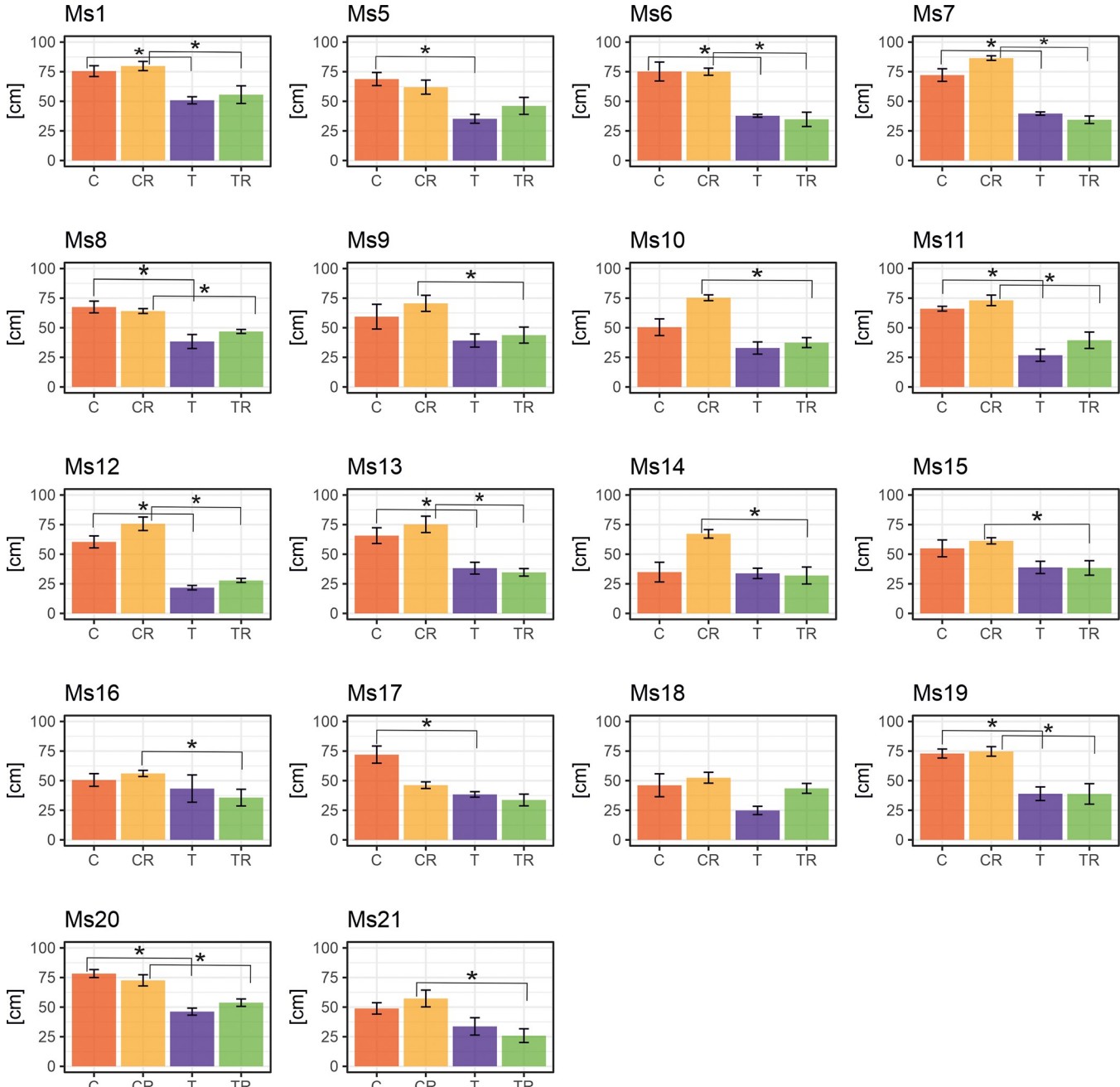

**Fig 2. Variation in response to chilling of the leaf length in 18 *M. sinensis* genotypes.** The data represent the mean values obtained from five biological replicates with error bars denoting the Standard Error of the Mean (SEM). Asterisks represent statistically significant differences between control, treatment, or recovery groups given by Welch's T-test with significance level α = 0.05. T–treatment-chilling stress; C–control for chilling treatment; TR—and treatment recovery; CR–control for treatment recovery.

decreased during chilling and recovery phases compared to control plants. While yield-related traits initially declined, they gradually increased during recovery, nearing control values, with a few exceptions.

Leaf length decreased across all *M. sinensis* genotypes during chilling, with the most significant declines in Ms12 (64%) and Ms11 (59%). Ms14 and Ms16, which grew fastest during chilling stress, showed smaller reductions of 3% and 14%, respectively, with Ms14 experiencing the

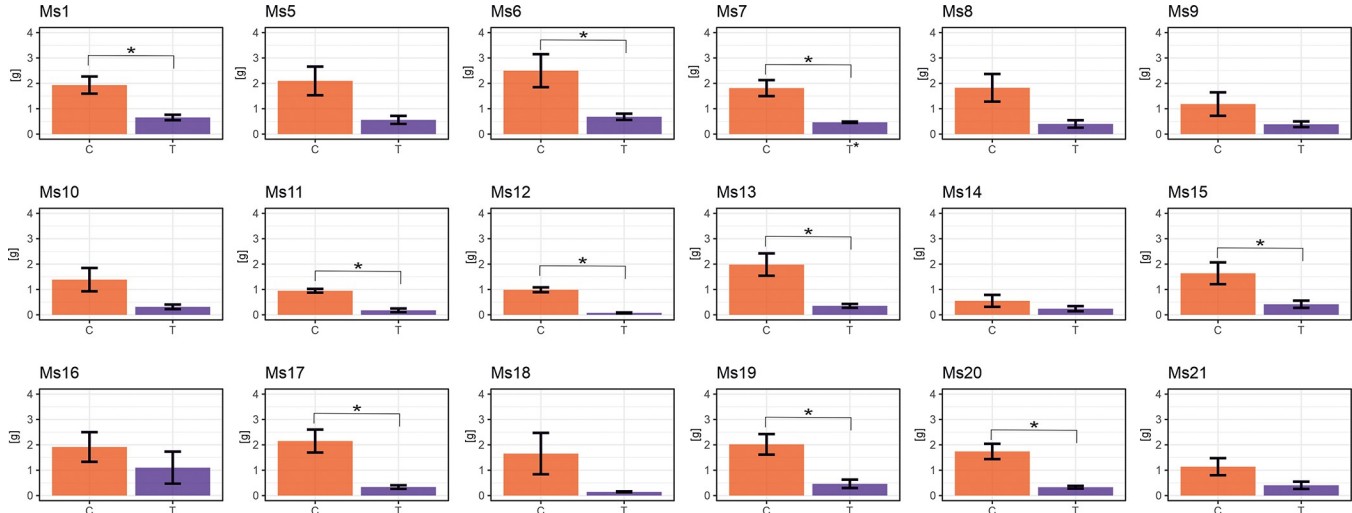

**Fig 3. Variation in response to chilling of the dry biomass yield in 18 *M. sinensis* genotypes.** The data represent the mean values obtained from five biological replicates with error bars denoting the Standard Error of the Mean (SEM). Asterisks represent statistically significant differences between control, treatment, or recovery groups given by Welch's T-test with significance level α = 0.05. T–treatment-chilling stress; C–control for chilling treatment; TR—and treatment recovery; CR–control for treatment recovery.

smallest reduction (3%) (Fig 2). Significant reductions in leaf length were observed in 11 genotypes, while seven genotypes showed no significant differences (Fig 2). Leaf length remained shorter across all genotypes during the recovery phase compared to control plants (Fig 2).

In terms of the number of leaves per stem during chilling, Ms12 and Ms18 exhibited the lowest values, with only four leaves per stem, while the highest values were observed in Ms16, with seven leaves per stem (S2 Fig).

During chilling stress, Ms12 and Ms18 had the fewest leaves per stem, while Ms16 had the most, followed by Ms6 and Ms5 (S2 Fig). Compared to these values relative to control plants, the smallest decreases in leaves per stem were seen in Ms16, Ms7, Ms11, and Ms14, while the largest decreases occurred in Ms18 and Ms12 (S2 Fig). Most genotypes showed a significant reduction in leaves per stem under chilling, in the majority of genotypes. During recovery, there were statistically significant differences in the number of leaves per stem compared to the control group, except for genotypes Ms18 and Ms19 (S2 Fig).

Regarding stem diameter under chilling, genotypes Ms18, Ms20 and Ms12 showed the largest stem diameter reductions (41%, 41% and 39%), while Ms21 (0%), Ms14 (5%) and Ms1 (8%) had the smallest reductions (S3 Fig). No significant reduction in the stem diameter when comparing plants subjected to chilling stress treatment with control plants in seven genotypes (S3 Fig). During recovery, there were no significant differences in the number of leaves per stem compared to controls in nine genotypes (S3 Fig).

Following the chilling treatment, the highest reductions in dry biomass compared to control values were observed in genotypes Ms12 (92%) and Ms18 (91%), while the smallest reduction was seen in Ms16 (42%) (Fig 3). Statistically significant biomass reductions were found in genotypes Ms1, Ms6, Ms7, Ms11, Ms12, Ms13, Ms15, Ms17, Ms19, and Ms20 (Fig 3). Conversely, the rest of the genotypes showed no statistically significant variations (Fig 3). These variations likely arise from genetic differences affecting tolerance and adaptation to chilling temperatures. Genotypes more susceptible to chilling, experienced significant biomass reduction. In contrast, genotypes exhibited resilience or adaptive mechanisms mitigating the chilling stress's negative effects on biomass accumulation.

## Genotypic variation in chlorophyll fluorescence parameters

During the chilling treatment, the $F_V/F_O$ ratio, a ratio between the rate constants of photochemical and non-photochemical deactivation of excited chlorophyll molecules [26], significantly decreased across all 18 *M. sinensis* genotypes, with reductions ranging from 36% (Ms14) to 71% (Ms1) of the control (S5 and S9 Figs). In the recovery phase, $F_V/F_O$ increased compared to the chilling treatment but remained below control levels. Notably, genotypes Ms1, Ms9, Ms18, and Ms21 showed significant differences in $F_V/F_O$ compared to the control plants (S5 and S9 Figs).

Chilling stress affected the quantum yield of primary photochemistry ($F_V/Fm$—φPo), which indicates the overall photosynthetic potential of active PSII reaction centers. A significant decrease in $F_V/F_m$ was observed, ranging from 21% (Ms9) to 55% (Ms15) compared to the control. This decline was notable for most genotypes, except for Ms9 and Ms13 (S6 and S9 Figs).

During the recovery phase, $F_V/F_m$ increased compared to the chilling treatment but still remained slightly lower than in the control, e.g., Ms6, Ms7, Ms11 and Ms12 (S6 and S9 Figs). These parameters ($F_v/F_o$ and $F_v/F_m$) exhibited a strong correlation (S1 Fig), however they did not react to chilling stress with the same level of sensitivity as $PI_{ABS}$ (Fig 4 and S9 Fig). To assess the impact of chilling stress and subsequent recovery (Fig 1A) on overall photosynthetic performance, we evaluated the Performance Index on Absorption Basis ($PI_{ABS}$). Chilling stress significantly reduced $PI_{ABS}$ across all genotypes, except for Ms14, for which $PI_{ABS}$ were still reduced but not significantly p-value = 0.07> α = 0.05). The lowest $PI_{ABS}$ was observed in Ms13, while Ms14, Ms15, and Ms16 had the highest values (Fig 4). Reductions in $PI_{ABS}$ ranged from 38% (Ms14) to 94% (Ms7 and Ms13) compared to the control (Fig 4 and S9 Fig). During the recovery phase, $PI_{ABS}$ increased but remained below control levels. Specifically, for four genotypes (Ms6, Ms7, Ms9 and Ms21), $PI_{ABS}$ differences were significantly lower compared to the control plants (Fig 4 and S9 Fig). In the chilling stress treatment, there was a notable increase in the heat dissipation of the PSII antenna, as indicated by the $F_o/F_m$ parameter, across all genotypes except Ms1 compared to control plants. These changes were statistically significant for all genotypes except Ms13 and Ms15 (Fig 5 and S9 Fig). During the recovery phase, $F_o/F_m$ decreased compared to the chilling treatment but remained similar to what was observed in the control group. The only exception was $F_o/F_m$ in Ms18, which was significantly higher compared to control plants (Fig 5 and S9 Fig).

During the chilling treatment, SPAD values, indicating chlorophyll content, decreased in all genotypes, ranging from 33% (Ms11) to 41% (Ms9, Ms14, Ms16) (S7 and S9 Figs), with 10 genotypes showing a significant difference compared to the control (Ms1, Ms5, Ms6, Ms7, Ms8, Ms11, Ms12, Ms13, Ms18, and Ms19). However, in the recovery phase, SPAD values increased relative to the chilling treatment but still remained lower than those in the control group in seven genotypes (S7 and S9 Figs). According to PCA analysis the maximum quantum yield of PSII ($F_v/F_m$) was strongly positively correlated with the ratio between the rate constants of photochemical and non-photochemical quenching of excited chlorophyll molecules ($F_v/F_o$) while negatively correlated with quantum yield of energy dissipation ($F_o/F_m$) located in the opposite direction of the biplot (Fig 1B), which was also confirmed by their correlations (S1 Fig).

## Genotypic variation in ratio of leaf elongation

To assess the impact of chilling-induced stress on leaf growth, we used the Leaf Elongation Rate (LER) alongside control plants and employed the Relative LER to compare each genotype to the control group. Among the 18 genotypes, we identified two subsets. The first group, consisting of 10 genotypes, showed LER values not significantly different from the control plants.

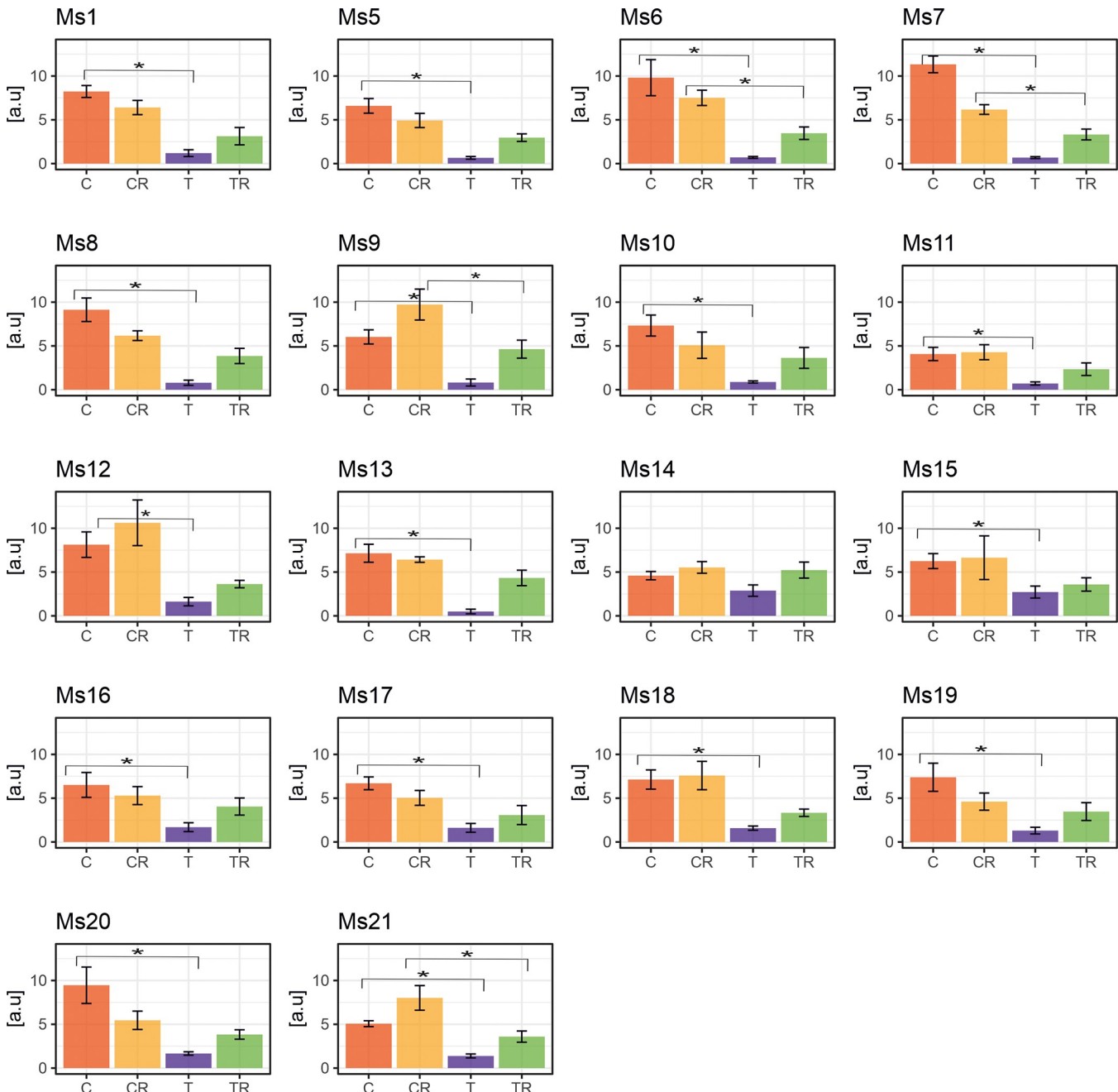

**Fig 4. Variation in response to chilling of the Performance Index on Absorption Basis (PI$_{ABS}$) fluorescence parameter in 18 *M. sinensis* genotypes.** The data represent the mean values obtained from five biological replicates with error bars denoting the Standard Error of the Mean (SEM). Asterisks represent statistically significant differences between control, treatment, or recovery groups given by Welch's T-test with significance level α = 0.05. T–treatment-chilling stress; C–control for chilling treatment; TR—and treatment recovery; CR–control for treatment recovery.

The second group, including Ms19, Ms13, Ms21, Ms17, Ms12, Ms11, Ms20, and Ms9, displayed a significant reduction in leaf elongation (α = 0.05; Fig 6A). Notably, three genotypes—Ms1, Ms16, and Ms10—exhibited a lower decrease in leaf elongation under chilling stress compared to the other 13 genotypes, achieving 60%, 52%, and 40% of the control growth, respectively (Fig 6A). Conversely, the genotypes Ms18, Ms19, Ms13 and Ms21 achieved the lowest relative LER at 5% 7%, 8%, and 9% of the control, respectively (Fig 6A).

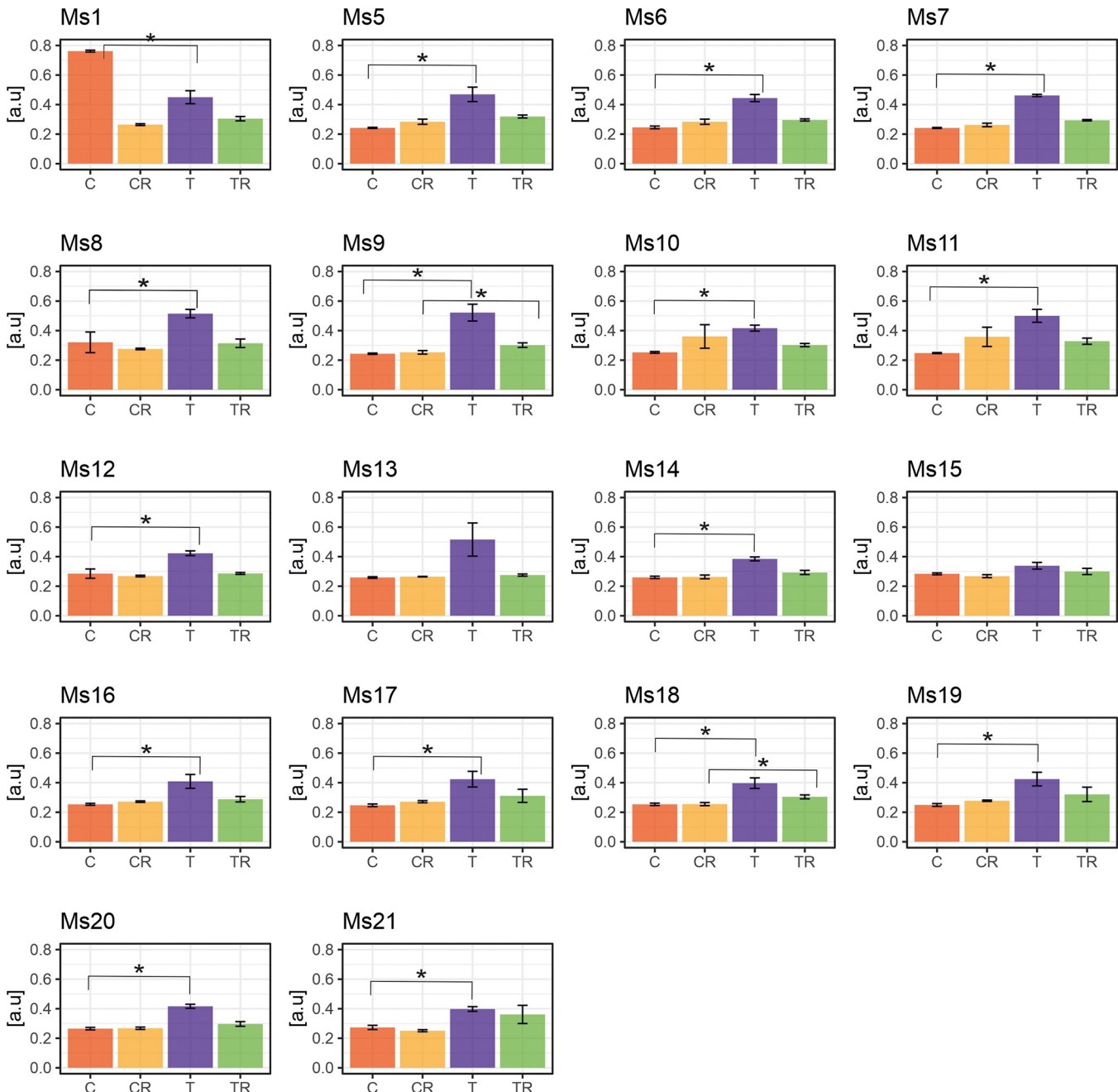

**Fig 5. Variation in response to chilling of the heat dissipation of the PSII antenna ($F_o/F_m$) fluorescence parameter in 18 *M. sinensis* genotypes.** The data represent the mean values obtained from five biological replicates with error bars denoting the Standard Error of the Mean (SEM). Asterisks represent statistically significant differences between control, treatment, or recovery groups given by Welch's T-test with significance level α = 0.05. T–treatment-chilling stress; C–control for chilling treatment; TR—and treatment recovery; CR–control for treatment recovery.

## The total Chilling Stress Response Index (SRI)

To identify genotypes with varying degrees of tolerance to chilling stress, we computed the Comprehensive Total Chilling Stress Response Index (SRI), integrating all observations from our research. The SRI identified a distinct cluster of genotypes—Ms16, Ms14, and Ms9—that displayed strong chilling tolerance, with Ms16 showing the highest tolerance. Conversely,

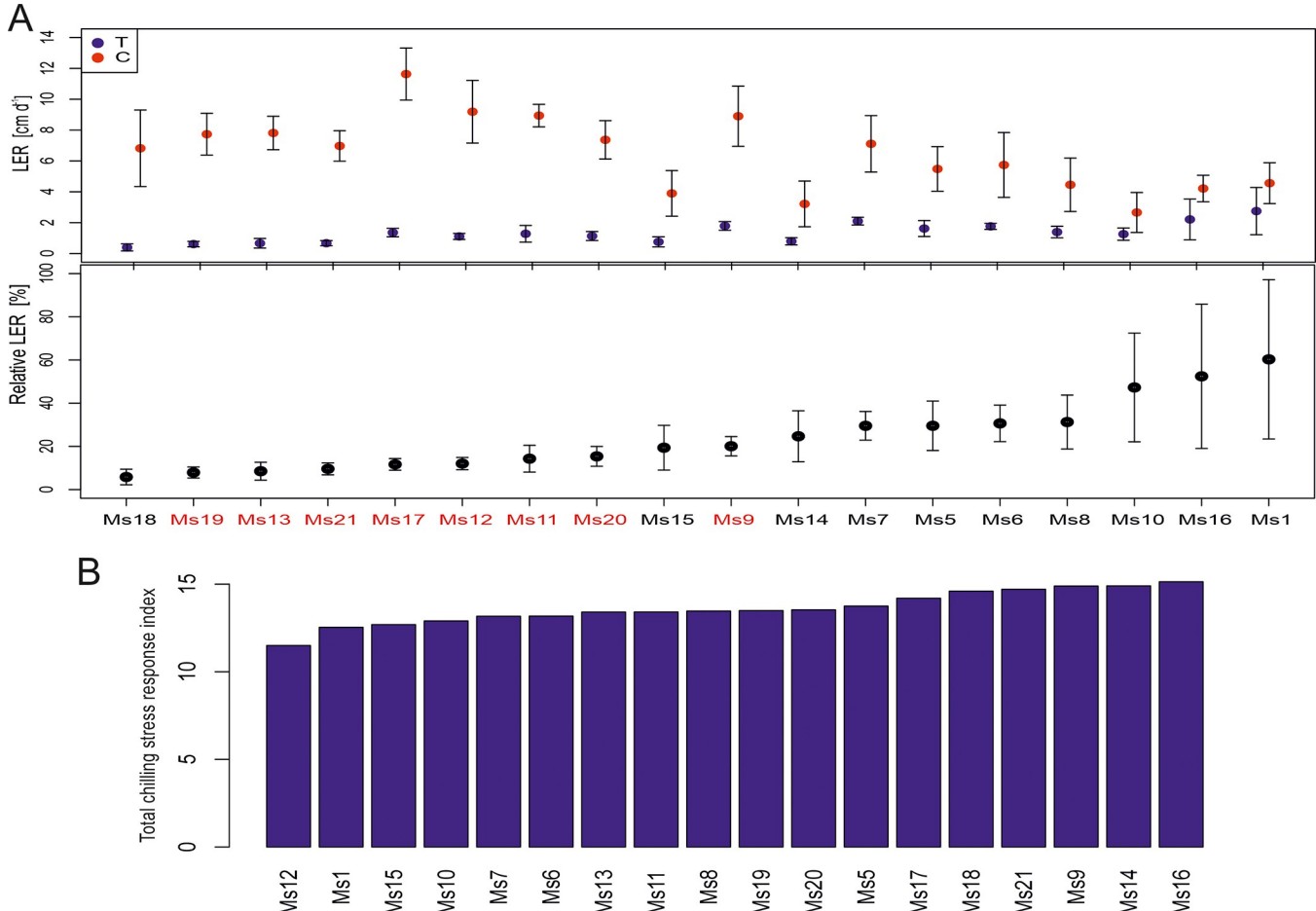

**Fig 6. A.** Leaf elongation rate at chilling temperature (T, blue) and control conditions (C, red) for 18 *M. sinensis* genotypes. Data are mean ±SE (n = 5 biological replicates). Red genotype names indicate significant difference between LERs in chilling and control conditions based on Welch's T-test with significance level α = 0.05; **B.** The total Chilling Stress Response Index (SRI) of 18 genotypes of *M. sinensis*.

Ms12 was identified as having comparatively lower chilling tolerance (Fig 6B). The remaining genotypes exhibited a relatively uniform response to chilling-induced stress (Fig 6B). Additionally, the mean Stress Susceptibility Index (SSI) confirmed the SRI results, highlighting Ms14 and Ms16 as the genotypes with the highest chilling tolerance, while Ms21 and Ms12 had the lowest tolerance among the studied genotypes (S8 Fig).

## Discussion

### What is the range of chilling tolerance differences that exists among the 18 *M. sinensis* genotypes?

*M. sinensis* has a wide distribution in East Asia [27, 28] primarily found in China, Korea, and Japan, extending as far north as Sakhalin and south to the Indochinese peninsula. This distribution overlaps with *M. sacchariflorus*, which is more common in northern regions, while *M. sinensis* predominates in the south [29]. Among the 18 *M. sinensis* genotypes examined, Ms12 shows increased susceptibility to chilling stress, whereas Ms16 exhibits the highest tolerance (Fig 6B). To understand the interactions between plant physiology, morphology, and biomass yield under chilling stress and during recovery, we employed Principal Component Analysis

(PCA) (Fig 1B). This method has proven effective in identifying key traits related to stress tolerance in various crops such as canola, rice, and maize [30–34]. In our research, PCA highlighted essential variables including dry biomass, leaf length, and stem diameter, which varied significantly among the genotypes (Fig 1B). For instance, dry biomass (Fig 3) ranged from 7.7% in Ms12 to 57.5% in Ms16 under chilling conditions relative to control, while leaf length (Fig 2) varied from 36% in Ms12 to 96.9% in Ms14. Stem diameter (S3 Fig) also showed substantial variation, from 58.5% in Ms18 to 100% in Ms21 (expressed as a percentage of control).

During chilling stress, the genotypes exhibited less variation compared to the control group, which allowed us to identify a subgroup (Ms12, Ms18, and Ms16) with unique responses. The control group demonstrated phenotypic plasticity, particularly in Ms14 and Ms1. Recovery periods saw increased variability among genotypes, with some showing distinct responses to the chilling conditions (Fig 1B). For example, leaf length during recovery ranged from 36.8% in Ms12 to 82.9% in Ms18 (expressed as a percentage of control, Fig 2), and stem diameter varied from 62.2% in Ms13 to 112.4% in Ms21 (relative to control, S3 Fig). Chlorophyll fluorescence parameters, such as $PI_{ABS}$, also varied, indicating differences in recovery efficiency among genotypes (Fig 4, S9 Fig).

The Leaf Elongation Rate (LER) [12] further illustrated these differences, with genotypes like Ms1, Ms16, and Ms10 showing the highest relative LER, whereas Ms18, Ms19, Ms13, and Ms21 had the lowest (Fig 6A). Previous studies comparing leaf elongation rates among various *M.* x *giganteus* and a total of 21 *M. sinensis* accessions along with four *M. sacchariflorus* accessions at low temperatures highlighted the significant range in chilling tolerance within the species [16].

The Total Chilling Stress Response Index (SRI) we developed (Fig 6B), which integrates biomass yield, chlorophyll fluorescence, and LER (Fig 6A), proved effective in assessing chilling tolerance. Tolerance indices have previously been applied successfully to determine genotypes and traits that possess significant capacity to endure abiotic stress conditions in wheat and rice [23, 35, 36]. Genotypes like Ms16, Ms14, and Ms9 demonstrated high tolerance, while Ms12 showed the lowest (Fig 6B). The SRI allowed us to assess chilling tolerance comprehensively, considering both biomass yield and growth performance under chilling conditions. This approach helps eliminate the trade-offs between yield potential and chilling tolerance during genotype selection, which is often a challenge when relying solely on the LER index [16].

## What distinct physiological attributes and reactions explain the chilling-tolerant variants from the chilling-sensitive ones within studied genotypes?

Our findings emphasize the importance of considering the intricate physiological responses, alongside phenotypic evaluations, and biomass yield, to comprehensively assess chilling tolerance across the 18 *M. sinensis* genotypes. Therefore, incorporating the SRI stress factor in our findings, along with PCA results, is crucial for selecting genotypes with both low and high chilling tolerance. Our study rejects the notion that a single parameter can be used to estimate the chilling tolerance of *M. sinensis*. Nonetheless, we found that $PI_{ABS}$ was a more sensitive parameter during the chilling treatment compared to $F_v/F_m$ and $F_v/F_o$ (Fig 4 and S9 Fig). Previous studies primarily used the $F_V/F_m$ ratio to assess chilling tolerance, but it is often insufficient [37]. Strasser et al. (2004) [38] introduced the performance index ($PI_{ABS}$), considering two key steps in photosynthesis. Soybean studies indicate a positive correlation between dark chilling effects on $PI_{ABS}$ and various photosynthetic factors. The Snop genotype displayed the least sensitivity to chilling, experiencing an average $PI_{ABS}$ reduction of 13%, while the highly

chilling-sensitive PAN809 genotype showed an average reduction of 65% [39]. Genotypes show varying dark chilling tolerance, with the $PI_{ABS}$ proving to be more effective than the $F_V/F_m$ in evaluating genotypes [39]. In our experiment, $PI_{ABS}$ proved to be the most effective parameter for detecting chilling stress effects. PCA and matrix correlation analyses revealed correlations between yield-related traits and dry biomass content, with leaf length showing the correlation (Fig 1B and S9 Fig). Throughout chilling and recovery, traits such as leaf length (Fig 2), number of leaves per stem (S2 Fig), stem diameter (S3 Fig), and shoot height (S4 Fig) exhibited declines compared to the control, gradually recovering in the later phase.

## How do two extreme genotypes react to chilling-induced stress?

In assessing the maximum primary yield of photochemistry, $F_v/F_o$ is a parameter that considers concurrent changes in both $F_m$ and $F_o$ [40]. In the chilling experiment, Ms16 exhibited a lesser reduction, whereas Ms12 demonstrated a more pronounced reduction in the $F_v/F_o$ (ratio between the rate constants of photochemical and non-photochemical quenching of excited chlorophyll molecules) (S5 and S9 Figs) [26] parameter compared to the control. The decreased $F_v/F_o$ values in fronds exposed to chilling stress indicate changes in the electron transport rate to the primary electron acceptors from PSII, along with a reduction in the number and size of the reaction centers. Various plant species have shown this decrease in the $F_v/F_o$ ratio due to environmental stress [26]. The altered $F_v/F_o$ may result from changes in minimum fluorescence in the dark ($F_o$), which affects the energy transfer from the antenna complex to the reaction center [41].

However, genotypes had similar reductions in maximum quantum yield of PSII ($F_v/F_m$) during chilling, with Ms16 at 45% and Ms12 at 42% compared to the control condition (S6 and S9 Figs). This confirms that utilizing this parameter is not a reliable indicator of chilling tolerance level [39, 42]. The performance index for PSII ($PI_{ABS}$) proved to be the most susceptible to chilling in both genotypes (Fig 4 and S9 Fig). $PI_{ABS}$ evaluates key processes related to photosynthetic activity, involving the absorption of light energy, excitation energy capture, and its conversion into electron transport within a PSII reaction center complex [42]. During the recovery period, the reduction remained relatively high for Ms12 at 65%, while for Ms16, it was 23%, indicating that Ms16 was closer to control values. This observation could imply that in the case of Ms12, there might have been partial damage to PSII, which became noticeable after the chilling stress period had ended. Consequently with Ms16 being more chilling tolerant than Ms12, during the chilling treatment the phase quantum yield of energy dissipation $F_o/F_m$ [43], displayed higher values for Ms16 compared to the control group, in contrast to Ms12 (Fig 5 and S9 Fig). This is supported by the observation that $F_o/F_m$, acts as a sensitive marker for stress conditions, showing significant increases under stress, which indicates engagement in non-photochemical quenching processes to protect PSII [44]. However, during the recovery period, both genotypes exhibited similar $F_o/F_m$ values, suggesting resilience and stabilization of energy dissipation mechanisms once stress is alleviated [44–46]. This suggests that Ms16 may have more effective mechanisms for protecting PSII through non-photochemical quenching processes [13, 47, 48]. In the chilling and recovery treatment, Ms16 had a smaller reduction in SPAD values, whereas Ms12 showed a greater decrease compared to the control plants (S7 and S9 Figs). This indicates that the higher chilling tolerant Ms16 can protect chlorophyll from degradation more effectively during chilling stress. This enhanced protection not only safeguards the chlorophyll but also means that more energy can be harvested by the higher chilling tolerant *M. sinensis*. This increased energy harvest must also be efficiently utilized photochemically to avoid photodamage. That might partially explain why during the chilling treatment, Ms16 exhibited a non-significant reduction in leaf length, whereas

Ms12 demonstrated a significant decrease compared to the control (Fig 2). Additionally, in terms of leaves per stem, Ms16 had the highest value, while Ms12 showed a considerable decrease during the chilling treatment compared to the control (S2 Fig). After the chilling treatment, the dry biomass exhibited more pronounced differences from the control values, with Ms16 showing the smallest reduction and Ms12 showing a significant decrease (Fig 3). Two of the most extreme genotypes in response to chilling were Ms12 (low chilling tolerance, SRI = 11.5) and Ms16 (high chilling tolerance, SRI = 15.1) (Fig 6B). All these measurements confirm or agree with direct measurements of chilling tolerance, such as growth under low temperature.

## Conclusions

This study suggests that the proposed tool-stress index is highly effective for the non-invasive, cost-efficient assessment of new germplasm's suitability for developing chilling-tolerant cereal and energy crop varieties. Chilling tolerance is an increasingly critical trait for future crop development, given that climate change leads to longer springs with more unpredictable temperature drops. To capitalize on spring soil moisture and the extended growing season, breeding more chilling-tolerant crops is essential. The observed genotypic variation and plasticity among the 18 *M. sinensis* genotypes during chilling treatment and recovery phases offer valuable insights. Coupled with advancements in high-throughput genome editing techniques, this observation presents an opportunity for conducting high-throughput screening. This approach holds the potential for unraveling the genetic mechanisms underlying adaptation to chilling stress. Such endeavors aim to deepen our understanding of tolerance mechanisms and potentially introduce a novel crop cultivar with enhanced chilling tolerance, paving the way for sustainable food and biomass production in challenging environmental conditions.

## Supporting information

**S1 Fig. Correlations between considered traits.**
(TIF)

**S2 Fig. Variation in response to chilling of number of leaves per stem in 18 *M. sinensis* genotypes.** The bars represent the mean ± Standard Error of the Mean (SEM; n = 5 biological replicates). Asterisks represent statistically significant differences between control, treatment, or recovery groups in Welch's T-test with significance level α = 0.05. T–treatment-chilling stress; C–control for chilling treatment; TR—treatment recovery; CR–control for treatment recovery.
(TIF)

**S3 Fig. Variation in response to chilling of stem diameter in 18 *M. sinensis* genotypes.** The bars represent the mean ± Standard Error of the Mean (SEM; n = 5 biological replicates). Asterisks represent statistically significant differences between control, treatment, or recovery groups in Welch's T-test with significance level α = 0.05. T–treatment-chilling stress; C–control for chilling treatment; TR—treatment recovery; CR–control for treatment recovery.
(TIF)

**S4 Fig. Variation in response to chilling of shoot height in 18 *M. sinensis* genotypes.** The bars represent the mean ± Standard Error of the Mean (SEM; n = 5 biological replicates). Asterisks represent statistically significant differences between control, treatment, or recovery groups in Welch's T-test with significance level α = 0.05. T–treatment-chilling stress; C–control for chilling treatment; TR—treatment recovery; CR–control for treatment recovery.
(TIF)

**S5 Fig. Variation in response to chilling of $F_v/F_o$ fluorescence parameter in 18 *M. sinensis* genotypes.** The data represent the mean values obtained from five biological replicates. Each column is accompanied by error bars denoting the Standard Error of the Mean (SEM). Asterisks represent statistically significant differences between control, treatment, or recovery groups given by Welch's T-test with significance level $\alpha = 0.05$. T–treatment-chilling stress; C–control for chilling treatment; TR—and treatment recovery; CR–control for treatment recovery.
(TIF)

**S6 Fig. Variation in response to chilling of $F_v/F_m$ fluorescence parameter in 18 *M. sinensis* genotypes.** The data represent the mean values obtained from five biological replicates. Each column is accompanied by error bars denoting the Standard Error of the Mean (SEM). Asterisks represent statistically significant differences between control, treatment, or recovery groups given by Welch's T-test with significance level $\alpha = 0.05$. T–treatment-chilling stress; C–control for chilling treatment; TR—and treatment recovery; CR–control for treatment recovery.
(TIF)

**S7 Fig. Variation in response to chilling of the SPAD-value in 18 *M. sinensis* genotypes.** The data represent the mean values obtained from five biological replicates. Each column is accompanied by error bars denoting the Standard Error of the Mean (SEM). Asterisks represent statistically significant differences between control and treatment, or recovery groups given by Welch's T-test with significance level $\alpha = 0.05$. T–treatment-chilling stress; C–control for chilling treatment; TR—and treatment recovery; CR–control for treatment recovery.
(TIF)

**S8 Fig. The mean Stress Susceptibility Index (SSI) of 18 genotypes of *M. sinensis*.**
(TIF)

**S9 Fig. Radar plots for photosynthetic parameters for treatment and recovery phase.**
(TIF)

**S1 Dataset. This file contains the data set used in the study.** The data points used to generate the graphs presented in the figures include all relevant variables and measurements.
(XLSX)

## Acknowledgments

The authors are thankful to Magdalena Tomaszewska, Aleksandra Kulka, Julianna Muszyńska and Agata Klepacka for their help in data collection and plant care.

## Author Contributions

**Conceptualization:** Karolina Sobańska, Martyna Przewoźnik.

**Data curation:** Karolina Sobańska, Monika Mokrzycka, Tomasz Pniewski.

**Formal analysis:** Monika Mokrzycka.

**Funding acquisition:** Karolina Sobańska, Tomasz Pniewski.

**Investigation:** Karolina Sobańska, Martyna Przewoźnik.

**Methodology:** Karolina Sobańska, Katarzyna Głowacka.

**Project administration:** Karolina Sobańska.

**Resources:** Karolina Sobańska, Monika Mokrzycka, Martyna Przewoźnik.

**Software:** Karolina Sobańska, Monika Mokrzycka.

**Supervision:** Karolina Sobańska.

**Validation:** Karolina Sobańska, Monika Mokrzycka.

**Visualization:** Karolina Sobańska, Monika Mokrzycka, Martyna Przewoźnik.

**Writing – original draft:** Karolina Sobańska, Monika Mokrzycka, Katarzyna Głowacka.

**Writing – review & editing:** Karolina Sobańska, Monika Mokrzycka, Martyna Przewoźnik, Tomasz Pniewski, Katarzyna Głowacka.

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
