## [Decision Letter · Decision Letter 0]

20 May 2024

PONE-D-24-12204Exploring Chilling Stress and Recovery Dynamics in C4 perennial grass of Miscanthus sinensisPLOS ONE

Dear Dr. Sobańska,

Thank you for submitting your manuscript to PLOS ONE. After careful consideration, we feel that it has merit but does not fully meet PLOS ONE’s publication criteria as it currently stands. Therefore, we invite you to submit a revised version of the manuscript that addresses the points raised during the review process.

We look forward to receiving your revised manuscript.

Kind regards,

Ali Akbar Ghasemi-Soloklui, Ph.D

Academic Editor

PLOS ONE

Journal Requirements:

"This work was supported by the National Center Science Poland (NCN) [grant 218/29/N/NZ9/00854]. "

Reviewers' comments:

Reviewer's Responses to Questions

**Comments to the Author**

1. Is the manuscript technically sound, and do the data support the conclusions?

Reviewer #1: Yes

Reviewer #2: Yes

2. Has the statistical analysis been performed appropriately and rigorously? 

Reviewer #1: Yes

Reviewer #2: Yes

3. Have the authors made all data underlying the findings in their manuscript fully available?

Reviewer #1: Yes

Reviewer #2: Yes

4. Is the manuscript presented in an intelligible fashion and written in standard English?

Reviewer #1: Yes

Reviewer #2: Yes

5. Review Comments to the Author

Reviewer #1: The manuscript entitled "Exploring Chilling Stress and Recovery Dynamics in C4 Perennial Grass of Miscanthus sinensis," submitted by Sobańska et al., presents a comprehensive study aimed at understanding the response of different Miscanthus sinensis genotypes to chilling stress and their recovery dynamics. This research is crucial for enhancing the agricultural viability and biomass production of Miscanthus in cooler climates, given its potential as a sustainable bioenergy source. The study meticulously evaluates 18 genotypes for their growth, chlorophyll fluorescence, and overall chilling tolerance, introducing a novel Total Chill Stress Response Index (SRI) that integrates various physiological and growth parameters. This index aids in identifying genotypes with enhanced chilling tolerance, thereby contributing significantly to the development of more resilient bioenergy crops.

Clarification on Experimental Conditions:

Light and Chilling Stress Interaction: Please clarify under what light conditions (presence or absence of light) the plants were subjected to chilling stress. The impact of light on photosynthetic machinery, especially PSII under stress, is crucial and should be distinctly addressed to understand the pathways affected by these conditions.

Ambiguities in Text:

Lines 95-107: These lines are ambiguous and need clearer expression. Please specify the details mentioned in this section to enhance understanding.

Lines 241-263: Similar to the above, this section lacks clarity in the explanation provided. It would benefit from a more straightforward presentation of the information.

Light Intensity Parameters:

Lines 135-153: The manuscript mentions a light intensity of 750 µmol m^-2 s^-1, which is significantly higher than the typical 300-500 µmol m^-2 s^-1 for C3 and C4 plants. Was this choice based on prior studies, and could this high intensity influence photoinhibition alongside chilling stress? Please provide a rationale or references to support this decision.

Technical Corrections:

Line 193 (PIABS Notation): Ensure that all scientific terms and abbreviations are correctly written throughout the manuscript to maintain professional and technical accuracy. For instance, PIABS should be uniformly formatted.

Methodology Description:

Lines 201-226: The use of a mixed linear model in the methods section is mentioned but not elaborated upon. Could you provide more details on its application and rationale?

Also, the parameters such as SRI and related ones need more details and should be presented more clearly.

Data Visualization:

Results Section: A spider plot could be beneficial to visually contrast the most and least tolerant genotypes based on photosynthetic parameters. This would allow for an immediate, comparative visual representation of the data.

Streamlining the Results:

General Over-detailing: The results section is overly detailed, often repeating information that is also shown in figures. Aim to highlight key findings succinctly, focusing on significant results that advance the understanding of chilling stress responses.

Lines 373-382: This specific portion is too wordy. Simplify the text to focus on pivotal results and their implications, removing extraneous details.

Discussion Section Improvements:

Focus and Repetition: Avoid redundant presentation of data that was detailed in the results. Instead, focus on discussing the implications of these findings, the mechanisms involved, and how they relate to existing literature.

Lines 419-446 and 430-446: These sections repeat information and lack clear linkage to broader scientific concepts. They should be revised for clarity and relevance.

Lines 447-481: Ensure that all claims are supported by robust references and evidence. This is crucial for maintaining scientific credibility.

Conclusion and Terminology:

Lines 528-533 and 536-550: The conclusions drawn here appear speculative without sufficient empirical support. Clarify these assertions and ensure they are backed by the results presented.

Lines 559, 568-572: The mention of "molecular genetic techniques" and "high-throughput genome editing techniques" seems out of context as these were not part of the experimental approaches described. Either remove these terms or clarify if they refer to potential future work.

Technical and Editorial Corrections:

References and Formatting: The reference section requires thorough revision to ensure all citations are correctly formatted and relevant. Additionally, scientific names should be italicized, and abbreviations like CO2 should be consistently presented in the correct format.

Deepening the Analysis:

Consider including a correlation matrix of the parameters to discuss the physiological responses more deeply, especially in the most sensitive and tolerant genotypes. This could provide insights into the interconnected pathways affected by chilling stress.

General Suggestions:

Reduce verbosity across the Results and Discussion sections and sharpen the focus on mechanistic insights and physiological implications.

The impact of chilling stress on photosynthetic parameters, particularly relating to PSII and electron transport chains, should be more explicitly connected to the observed physiological responses. Also, shouls consider the fact that during the chilling stress, the combination of reactions can cause the stress which in ETC the pool of NADP+/NADPH or the capacity if the ETC should be consider. Just mentioning the NPQ reactions without any proof is not a reliable way!

These suggestions aim to enhance the manuscript's scientific rigor, clarity, and contribution to the field, ensuring that the study's results are communicated effectively and are supported by solid experimental evidence.

Reviewer #2: This research has considered appropriate assumptions. Data analysis is done correctly. The results are new and noteworthy, and it has also provided a suitable discussion. Therefore, its publication is recommended.

6. PLOS authors have the option to publish the peer review history of their article (what does this mean?). If published, this will include your full peer review and any attached files.

Reviewer #1: No

Reviewer #2: No

---

## [Author Response · Author response to Decision Letter 0]

24 Jun 2024

Reviewer's Responses to Questions

Comments to the Author

1. Is the manuscript technically sound, and do the data support the conclusions?

Reviewer #1: Yes

Reviewer #2: Yes

2. Has the statistical analysis been performed appropriately and rigorously? 

Reviewer #1: Yes

Reviewer #2: Yes

3. Have the authors made all data underlying the findings in their manuscript fully available?

Reviewer #1: Yes

Reviewer #2: Yes

4. Is the manuscript presented in an intelligible fashion and written in standard English?

Reviewer #1: Yes

Reviewer #2: Yes

5. Review Comments to the Author

Reviewer #1: 

The manuscript entitled "Exploring Chilling Stress and Recovery Dynamics in C4 Perennial Grass of Miscanthus sinensis," submitted by Sobańska et al., presents a comprehensive study aimed at understanding the response of different Miscanthus sinensis genotypes to chilling stress and their recovery dynamics. This research is crucial for enhancing the agricultural viability and biomass production of Miscanthus in cooler climates, given its potential as a sustainable bioenergy source. The study meticulously evaluates 18 genotypes for their growth, chlorophyll fluorescence, and overall chilling tolerance, introducing a novel Total Chill Stress Response Index (SRI) that integrates various physiological and growth parameters. This index aids in identifying genotypes with enhanced chilling tolerance, thereby contributing significantly to the development of more resilient bioenergy crops.

We would like to sincerely thank Reviewer #1 for the valuable comments on our work and positive assessment of our work.

Clarification on Experimental Conditions:

Light and Chilling Stress Interaction: Please clarify under what light conditions (presence or absence of light) the plants were subjected to chilling stress. The impact of light on photosynthetic machinery, especially PSII under stress, is crucial and should be distinctly addressed to understand the pathways affected by these conditions.

A: The plants were grown under the light conditions described in the Materials and Methods section and in Fig 1A. Specifically “They were grown in a greenhouse under white fluorescent tubes, providing a 12-hour photoperiod of photosynthetically active radiation at 750 µmol m-2 s-1 light intensity at canopy level, with temperatures maintained at 25°C during the day and 19°C at night, and relative humidity at 60% ± 10%” (Fig. 1A) (Revision manuscript with track changes).

Ambiguities in Text:

Lines 95-107: These lines are ambiguous and need clearer expression. Please specify the details mentioned in this section to enhance understanding.

A: Thank you for this comment. We agree that the text lacked clarity. The section has been rewritten in the revision manuscript with track changes version (Line 111-131). 

Lines 241-263: Similar to the above, this section lacks clarity in the explanation provided. It would benefit from a more straightforward presentation of the information.

A: Thank you for this comment. We agree that the text lacked clarity. The section has been rewritten in the revision manuscript with track changes version (Line 282- 312 , Revision manuscript with track changes).

Light Intensity Parameters:

Lines 135-153: The manuscript mentions a light intensity of 750 µmol m^-2 s^-1, which is significantly higher than the typical 300-500 µmol m^-2 s^-1 for C3 and C4 plants. Was this choice based on prior studies, and could this high intensity influence photoinhibition alongside chilling stress? Please provide a rationale or references to support this decision.

Chilling tolerance is most critical in spring and early fall when the combination of high light with chilling causes photoinhibition. Furthermore, the light conditions in that leaves develop are strictly related to photosynthetic capacity via adjustment of chlorophyll content and stoichiometry of antenna complexes to light conditions. To perform our experiment in the light conditions which field-grown plants might experience in spring and early fall we chose 750 µmol m-2 s-1. 

The new text has been added to the Material and Methods section clarifying the choice of light conditions: “The experiment was performed under 750 µmol m-2 s-1 of light to mimic the light conditions which field-grown plants might experience in spring and early fall when chilling events occur.” (Line 163-165, Revision manuscript with track changes).

Technical Corrections:

Line 193 (PIABS Notation): Ensure that all scientific terms and abbreviations are correctly written throughout the manuscript to maintain professional and technical accuracy. For instance, PIABS should be uniformly formatted.

A: Thank you for this comment. In the revised version of the manuscript, we made sure that the scientific terms and abbreviations are correctly written and uniformly formatted throughout the manuscript. (Revision manuscript with track changes).

Methodology Description:

Lines 201-226: The use of a mixed linear model in the methods section is mentioned but not elaborated upon. Could you provide more details on its application and rationale?

Also, the parameters such as SRI and related ones need more details and should be presented more clearly.

A: Thank you for this comment. Methodology description has been extended by adding clarification, rationale and formulas. In response to this comment the new text has been added to the revision with track changes version of the manuscript in the lines 247-250, 261-263.

Data Visualization:

Results Section: A spider plot could be beneficial to visually contrast the most and least tolerant genotypes based on photosynthetic parameters. This would allow for an immediate, comparative visual representation of the data.

A: Thank you for this suggestion. The radar plots for relative photosynthetic parameters for treatment and recovery have been added to the revised version of the manuscript (S9 Fig). The section has been rewritten to make clearer the reference to future work (Lines: 408, 412, 417, 421, 423, 433, 438,444, 447, 450, 456, 586, 613, 625) (Revision manuscript with track changes).

Streamlining the Results:

General Over-detailing: The results section is overly detailed, often repeating information that is also shown in figures. Aim to highlight key findings succinctly, focusing on significant results that advance the understanding of chilling stress responses.

Lines 373-382: This specific portion is too wordy. Simplify the text to focus on pivotal results and their implications, removing extraneous details.

A: Thank you for this comment. We trimmed and rewrote the text to concisely highlight key findings and focus on significant results. (Line 269-491, Revision manuscript with track changes).

Discussion Section Improvements:

Focus and Repetition: Avoid redundant presentation of data that was detailed in the results. Instead, focus on discussing the implications of these findings, the mechanisms involved, and how they relate to existing literature.

Lines 419-446 and 430-446: These sections repeat information and lack clear linkage to broader scientific concepts. They should be revised for clarity and relevance.

Lines 447-481: Ensure that all claims are supported by robust references and evidence. This is crucial for maintaining scientific credibility.

A: Thank you for this comment. We have trimmed and rewrote the text to focus on the implications of our findings, link them with broader scientific concepts and add references to evidence when applicable (Lines 496-749, Revision manuscript with track changes).

Conclusion and Terminology:

Lines 528-533 and 536-550: The conclusions drawn here appear speculative without sufficient empirical support. Clarify these assertions and ensure they are backed by the results presented.

A: Thank you for this comment. The text has been rewritten in the revision with track changes of the manuscript (Line 752 - 766).

Lines 559, 568-572: The mention of "molecular genetic techniques" and "high-throughput genome editing techniques" seems out of context as these were not part of the experimental approaches described. Either remove these terms or clarify if they refer to potential future work.

A: This section was intended to present new perspectives in the next research on chilling stress resulting from the selection of genotypes with varying tolerance to cold, including the application of high-throughput genome editing techniques. The section has been rewritten to make clearer the reference to future work (Line 757 - 762, Revision manuscript with track changes). 

Technical and Editorial Corrections:

References and Formatting: The reference section requires thorough revision to ensure all citations are correctly formatted and relevant. Additionally, scientific names should be italicized, and abbreviations like CO2 should be consistently presented in the correct format.

A: We apologize for the mistakes in formatting the reference list. The list has been corrected in the revised version of the manuscript (Line: 788-954, Revision manuscript with track changes). 

Deepening the Analysis:

Consider including a correlation matrix of the parameters to discuss the physiological responses more deeply, especially in the most sensitive and tolerant genotypes. This could provide insights into the interconnected pathways affected by chilling stress.

A: Thank you for this suggestion. We have added a correlation matrix (S1 Fig) to the revised version of the manuscript which extends the basic information about the degree of correlation that was already presented on biplot (Line: 313-318; 421-422; 455-460, Revision manuscript with track changes).

General Suggestions:

Reduce verbosity across the Results and Discussion sections and sharpen the focus on mechanistic insights and physiological implications.

The impact of chilling stress on photosynthetic parameters, particularly relating to PSII and electron transport chains, should be more explicitly connected to the observed physiological responses. Also, shouls consider the fact that during the chilling stress, the combination of reactions can cause the stress which in ETC the pool of NADP+/NADPH or the capacity if the ETC should be consider. Just mentioning the NPQ reactions without any proof is not a reliable way! 

A: In this study, we mainly focused on developing a tool - the Total Chilling Stress Index (SRI) - that enables a comprehensive selection of genotypes that are more resistant to cold. The text of the Results and Discussion section has been substantially trimmed. We also re-worded the text to make a more explicit connection between measured parameters and plant physiological response (Lines 630-749, Revision manuscript with track changes).

Reviewer #2: This research has considered appropriate assumptions. Data analysis is done correctly. The results are new and noteworthy, and it has also provided a suitable discussion. Therefore, its publication is recommended.

Thank you for your recommendation.

6. PLOS authors have the option to publish the peer review history of their article (what does this mean?). If published, this will include your full peer review and any attached files.

Do you want your identity to be public for this peer review? For information about this choice, including consent withdrawal, please see our Privacy Policy.

Reviewer #1: No

Reviewer #2: No

---

## [Decision Letter · Decision Letter 1]

18 Jul 2024

Exploring Chilling Stress and Recovery Dynamics in C4 perennial grass of Miscanthus sinensis

PONE-D-24-12204R1

Dear Dr. Sobańska,

We’re pleased to inform you that your manuscript has been judged scientifically suitable for publication and will be formally accepted for publication once it meets all outstanding technical requirements.

Kind regards,

Ali Akbar Ghasemi-Soloklui, Ph.D

Academic Editor

PLOS ONE

Additional Editor Comments (optional):

Reviewers' comments:

Reviewer's Responses to Questions

**Comments to the Author**

1. If the authors have adequately addressed your comments raised in a previous round of review and you feel that this manuscript is now acceptable for publication, you may indicate that here to bypass the “Comments to the Author” section, enter your conflict of interest statement in the “Confidential to Editor” section, and submit your "Accept" recommendation.

Reviewer #1: All comments have been addressed

2. Is the manuscript technically sound, and do the data support the conclusions?

Reviewer #1: Yes

3. Has the statistical analysis been performed appropriately and rigorously? 

Reviewer #1: Yes

4. Have the authors made all data underlying the findings in their manuscript fully available?

Reviewer #1: Yes

5. Is the manuscript presented in an intelligible fashion and written in standard English?

Reviewer #1: Yes

6. Review Comments to the Author

Reviewer #1: I declare that I have no conflicts of interest that would influence my review and recommendation of this manuscript. My review has been conducted impartially, and my assessment is based solely on the quality and content of the manuscript.

Sincerely,

7. PLOS authors have the option to publish the peer review history of their article (what does this mean?). If published, this will include your full peer review and any attached files.

Reviewer #1: No

---

## [Editor Report · Acceptance letter]

2 Aug 2024

PONE-D-24-12204R1 

PLOS ONE

Dear Dr. Sobańska, 

I'm pleased to inform you that your manuscript has been deemed suitable for publication in PLOS ONE. Congratulations! Your manuscript is now being handed over to our production team.

Kind regards, 

on behalf of

Dr. Ali Akbar Ghasemi-Soloklui 

Academic Editor

PLOS ONE